# `ern`: An R package to estimate the effective reproduction number using clinical and wastewater surveillance data

David Champredon *, Irena Papst, Warsame Yusuf

Public Health Risk Sciences Division, Public Health Agency of Canada, National Microbiology Laboratory, Guelph, Ontario, Canada

* david.champredon@canada.ca

**Data Availability Statement:** The R code for the package is currently available on the official R repository CRAN (https://CRAN.R-project.org/package=ern) and GitHub: https://github.com/phac-nml-phrsd/ern The data used in this

## Abstract

The effective reproduction number, $\mathcal{R}_t$, is an important epidemiological metric used to assess the state of an epidemic, as well as the effectiveness of public health interventions undertaken in response. When $\mathcal{R}_t$ is above one, it indicates that new infections are increasing, and thus the epidemic is growing, while an $\mathcal{R}_t$ is below one indicates that new infections are decreasing, and so the epidemic is under control. There are several established software packages that are readily available to statistically estimate $\mathcal{R}_t$ using clinical surveillance data. However, there are comparatively few accessible tools for estimating $\mathcal{R}_t$ from pathogen wastewater concentration, a surveillance data stream that cemented its utility during the COVID-19 pandemic. We present the R package `ern` that aims to perform the estimation of the effective reproduction number from real-world wastewater or aggregated clinical surveillance data in a user-friendly way.

## Introduction

The effective reproduction number, commonly denoted as $\mathcal{R}_t$, is a key metric in epidemiology. It is defined as the average number of new infections generated by an infected individual at time $t$ during an epidemic. It differs from the basic reproduction number, $\mathcal{R}_0$, in that it additionally accounts for changes in population susceptibility and transmission at a given point in time. The parameter $\mathcal{R}_t$ effectively measures the strength of transmission of an infectious pathogen within a population [1]. The value of $\mathcal{R}_t$ has a simple interpretation depending on whether it is greater than, equal to, or less than one: it implies that the number of new infections is either increasing, constant, or decreasing over time, respectively. Usually, $\mathcal{R}_t$ is estimated using the daily number of new cases reported via clinical surveillance. The importance of $\mathcal{R}_t$ was reinforced during the SARS-CoV-2 pandemic when its estimates supported public health decisions in many jurisdictions worldwide [2].

Wastewater-based epidemiological surveillance emerged as a critical component of the public health arsenal to monitor the COVID-19 pandemic (*e.g.,* [3, 4]), despite being used since at least since the 1940s to monitor the poliovirus [5]. While individuals infected with SARS-CoV-2 shed viral particles through various routes (such as urine, saliva, and sputum),

manuscript is publicly available and attached to the package.

**Funding:** The author(s) received no specific funding for this work.

**Competing interests:** The authors have declared that no competing interests exist.

stool shedding is the dominant source of viral shedding when examining community-level wastewater surveillance [6]. Once shed, viral particles enter the sewer network and reside in wastewater. Wastewater samples are typically collected at treatment plants and viral RNA is extracted from these samples using various laboratory methods. The concentration of viral RNA in these samples can be quantified using real-time quantitative polymerase chain reaction (RT-qPCR) as well as digital droplet PCR. The concentration is assumed to be proportional to the infection prevalence in the community living in the catchment area (up to a conversion factor). Fecal shedding occurs passively and irrespective of the symptomatic status of the infected individual [7], although shedding is likely to be at its peak during the symptomatic period [8]. Hence wastewater surveillance data does not have the same biases as clinical surveillance data, which tends to focus on symptomatic/severe infections.

In light of the utility of using wastewater-based surveillance during the COVID-19 pandemic, this methodology has been applied successfully to several other pathogens: human influenza, respiratory syncytial virus, and mpox are now routinely monitored in wastewater samples in many jurisdictions [9, 10]. Therefore, it is important for the public health community to be able to easily estimate $\mathcal{R}_t$ of an infectious disease from wastewater data. Moreover, as wastewater-based epidemiological surveillance expands, public health organizations will likely leverage *both* clinical and wastewater-based surveillance data to monitor the spread of pathogens. As such, it would be useful to have a tool that estimates $\mathcal{R}_t$ concordantly across both of these data sources.

The literature on methods to estimate $\mathcal{R}_t$ from clinical data is vast due to the importance of $\mathcal{R}_t$ in infectious disease epidemiology (for example [1, 11–15]). On the contrary, few studies have attempted to estimate $\mathcal{R}_t$ from wastewater data. Huisman *et al.* [16] proposed a method based on deconvoluting the fecal shedding distribution. Previous work has developed epidemic compartmental models that can integrate wastewater-based surveillance [17–19] but $\mathcal{R}_t$ cannot be derived explicitly (except for [19]). Jiang *et al.* [20] derived $\mathcal{R}_t$ from an artificial neural network, and Amman *et al.* [21] approximated $\mathcal{R}_t$ of SARS-CoV-2 variants from their relative abundance in wastewater samples. While these methods are useful, there have been relatively few efforts to port these theoretical frameworks into user-friendly software to apply them to real-world wastewater data. One recently-released R package, `EpiSewer`, aims to address this gap [22].

Clinical data are often reported as aggregated cases over a period of time, typically weekly. However, a key parameter in estimating $\mathcal{R}_t$ is the distribution of the intrinsic generation interval (defined as the interval between the time when an individual is infected by an infector and the time when this infector was infected). For many infectious pathogens, this interval is on the order of days. Many existing implementations of $\mathcal{R}_t$ estimation in R libraries require that the input data (clinical case reports) and the specification of the intrinsic generation interval [23] are on the same timescale (*e.g.,* days). For example, H1N1 influenza has a mean intrinsic generation interval of about 3 days and a maximum value of about 7 days [24, 25]. If the data is reported weekly, it is not possible to define the generation interval distribution meaningfully in units of week. This is because the generation interval distribution must be discrete for existing methods, so it is not as easy as defining a continuous distribution rescaled to weeks. Hence, before estimating $\mathcal{R}_t$ with existing methods, the input data must first be disaggregated onto the scale of days, which is not a straightforward process.

Several R packages exist to estimate $\mathcal{R}_t$ from clinical data. One popular package is `EpiEstim`, which initially implemented a Poisson-based model of the renewal equation [26]. This package has recently been improved to handle aggregated input data [27]. Briefly, the approach to estimating $\mathcal{R}_t$ from aggregated clinical reports (typically reported weekly)

relies on an expectation-maximization algorithm to disaggregate the counts into daily case reports, assuming a local exponential growth for transmission. As a result of this assumption, the inferred daily case reports have a piecewise exponential form, which may be problematic for downstream applications. Moreover, `EpiEstim` does not explicitly handle the various time delays like, for example, incubation period and reporting delays (the time between symptoms onset and reporting of a case) typically encountered in practice with epidemiological reports.

`EpiNow2` is also a recent R package that aims to improve the estimation of $\mathcal{R}_t$ including for example reporting delays and periodicity, as well as the propagation of parameter uncertainty [28]. The package also provides tools for short-term forecasting of case reports but cannot handle explicitly non-daily (*e.g.,* weekly) reporting. Another R package, `epidemia` provides a regression-based framework to estimate $\mathcal{R}_t$ from daily clinical data [29]. We note that while theoretically possible, estimating $\mathcal{R}_t$ from wastewater data with `EpiNow2` or `epidemia` is not straightforward, especially for users who do not have a modelling background. Moreover, because of their reliance on the Bayesian inference software Stan [30], computing time may be long. The R package `estimateR` is another tool to estimate $\mathcal{R}_t$ from clinical data but does not explicitly handle wastewater data or aggregated clinical data [31].

Here, we present the R library `ern` to address the gaps identified above, specifically:

- to disaggregate the clinical reports into a shorter time unit to enable estimation of $\mathcal{R}_t$ using an intrinsic generation interval on a useful timescale;

- to provide a framework to estimate $\mathcal{R}_t$ from wastewater data, consistent with an estimation based on clinical data;

- to provide a user-friendly interface geared at public-health practitioners that may have limited proficiency in the R programming language;

- to perform an efficient and rapid $\mathcal{R}_t$ estimation.

Table 1 summarises key features of the R packages discussed above, along with the `ern` package.

The `ern` package ultimately uses the `EpiEstim` package for the core of the $\mathcal{R}_t$ computation as `EpiEstim` already provides a robust and one of the fastest implementations of well-tested $\mathcal{R}_t$ estimation algorithms. However, `ern` wraps complex and critical features for estimating $\mathcal{R}_t$ from real-world clinical and wastewater data that have not all been implemented in any one existing R package for $\mathcal{R}_t$ estimation.

**Table 1. A comparison of `ern` with other R packages built to estimate $\mathcal{R}_t$ from epidemiological data.** Checkmarks (✓) indicate the presence of a feature and crosses (×) indicate the absence. A cross with an asterisk (×*) denotes a feature not built-in the package but technically possible though not straightforward for the average user (*e.g.,* they may require additional modelling knowledge and/or the use of advanced/less documented features).

| R Package | Accepted inputs | | Available features | |
|---|---|---|---|---|
| | Wastewater concentration | Daily clinical case data | Reporting delays | Disaggregate case data |
| ern | ✓ | ✓ | ✓ | ✓ |
| EpiSewer | ✓ | × | ✓ | × |
| EpiEstim | × | ✓ | × | ✓ |
| EpiNow2 | ×* | ✓ | ✓ | ×* |
| epidemia | ×* | ✓ | ✓ | ×* |
| estimateR | ×* | ✓ | ✓ | × |

## Materials and methods

The R code for the `ern` package is available on the Comprehensive R Archive Network at https://cran.r-project.org/web/packages/ern/index.html.

Fig 1 gives a high-level overview of how the `ern` package computes $\mathcal{R}_t$ for both wastewater and clinical input data. The pipeline for each data stream has three components:

1. Estimating daily incidence from the raw data (wastewater or clinical)

2. Estimating $\mathcal{R}_t$ from the estimated daily incidence

3. Repeating $\mathcal{R}_t$ estimates (previous two components) to generate an ensemble reflecting various sources of uncertainty

Throughout this work, we use the term *incidence* to denote the "true" underlying incidence of infections, as opposed to reported incidence (from clinical data), which we instead refer to as *reports* or reported cases.

Dashed elements represent optional components. Layered boxes represent replicates from resampling that inform uncertainty in the final $\mathcal{R}_t$ ensemble. Resampled elements include the distributions used in deconvolutions and `EpiEstim` (sampled from the specified family of distributions for each quantity), the set of inferred daily reports (when these are estimated), and the underreporting proportion.

### Estimating daily incidence with wastewater data

Our approach to estimating the daily incidence time series from wastewater data is similar to the one taken in [16], where the concentration of pathogen shed in wastewater, $w_t$, is assumed to be the convolution of the incidence of infections, $i$, and the fecal shedding distribution $f$ (the relative proportion of pathogen shed in feces as a function of time since infection) of an

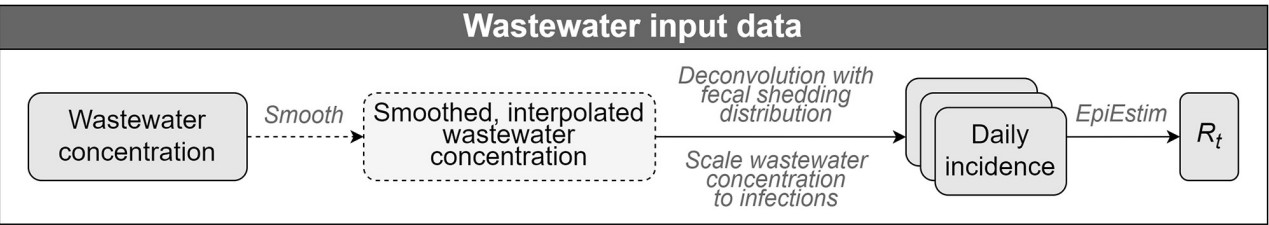

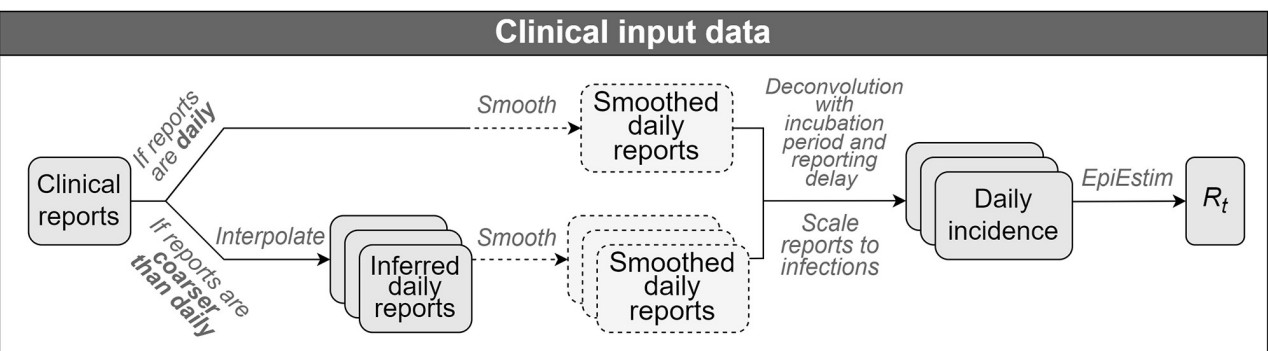

**Fig 1. Overview of the `ern` data pipeline to estimate $\mathcal{R}_t$.**

average infected individual:

$$w_t = \omega \sum_{k=1}^{t-1} i(t-k)f(k) \tag{1}$$

The function $f$ can be defined such that $\sum_{k>0} f(k) = 1$. The parameter $\omega$ denotes how much a single average infection contributes to wastewater concentration in total over the course of infection, as measured in the sewer system. This parameter captures baseline average shedding, but also reflects the loss of viral particles measured between the shedding and downstream sampling locations (dependant on the sewer system, environmental factors, and the processing pipeline of the laboratory).

Since we model the wastewater signal as a convolution of incidence with the fecal shedding distribution, we must perform a *deconvolution* of the wastewater signal with the fecal shedding distribution to recover incidence for $\mathcal{R}_t$ estimation. However, sampled pathogen concentration in wastewater tends to be a noisy signal, so we smooth the time series of concentrations $w_t$. Wastewater samples are often taken a few days a week, so smoothing additionally interpolates the signal on a daily scale, which is a requirement for working with an intrinsic generation interval measured in days to estimate $\mathcal{R}_t$. Hence, we obtain $W_t$, a *daily interpolated* and *smoothed* time series of pathogen concentrations in wastewater:

$$W_t = \text{smooth\_interpolation}(w_t, \theta) \tag{2}$$

where $\theta$ are the smoothing parameters.

The smoothing algorithms implemented in `ern` are moving average and LOESS (LOcally Estimated Scatterplot Smoothing), with default values set to provide a light smoothing of the time series. The moving average smooths a time series by taking an unweighted mean of all points in a window of each time point. `ern` users can specify the width and centering of the window with respect to the focal time point. The LOESS method is a generalisation of the moving average [32]; it still operates across subsets of the time series, but instead of computing the unweighted mean in each of these windows, it performs a weighted linear regression at each point and returns the predicted value of the focal time point. Weighting is done by distance from the focal time point, with closer points carrying more weight. The window size is controlled by a span parameter, which an `ern` user can specify, along with a minimum concentration to prevent zero or negative values in the smoothed time series when inputting low-concentration measurements.

Finally, to extract the daily incidence $i(t)$, we substitute $w_t$ by $W_t$ in Eq 1 and we use the Richardson-Lucy algorithm [33–35] to deconvolute $W_t$ using the fecal shedding distribution $f$ as the kernel:

$$i(t) = RL(W, f) \tag{3}$$

where $RL$ represents the deconvolution algorithm.

### Estimating daily incidence with clinical data

To estimate a daily incidence time series from *daily* clinical reports, the reports are optionally smoothed to eliminate some noise from the signal. As with wastewater input data, the smoothing algorithms available in `ern` are LOESS and moving average. Then, reports are scaled to account for underreporting and bring the signal to the scale of actual infections. Next, the smoothed and scaled time series is deconvoluted (similarly as in the wastewater method) using i) a reporting delay distribution kernel and ii) an incubation period

distribution kernel. These two deconvolutions estimate daily "true" incidence (*i.e.,* tallied by date of *infection*, not the *report* date).

In some cases, the clinical reporting frequency may not be compatible with the relevant timescale of the intrinsic generation interval distribution. For example, seasonal influenza cases are typically reported on a weekly basis, but its generation interval should be defined in units of days because it is shorter than a week for most cases [24, 25]. (For a detailed discussion on why the reporting frequency and timescale of the intrinsic interval must match, see section "Daily incidence to $\mathcal{R}_t$".) The package ern implements two methods to interpolate aggregate reports and produce inferred daily reports used to compute $\mathcal{R}_t$.

The first method is called the "renewal" method as it involves a statistical model that infers the latent daily reports from aggregate counts using a standard "Susceptible-Infectious-Recovered" (SIR) epidemic model via the renewal equation [36, 37].

This approach ensures the inferred daily reports follow a realistic epidemic curve, as opposed to, *e.g.,* an ad-hoc estimate such as naively dividing weekly reports by 7. A poor approximation of the exponential transmission process of the disease, as reflected in the inferred daily reports, could significantly impact the quality of the $\mathcal{R}_t$ estimates. See S1 File for an example.

With the renewal interpolation method, SIR model parameters are fitted to the aggregated (*e.g.,* weekly) clinical reports using a Markov Chain Monte Carlo (MCMC) algorithm and then daily reports are inferred from the fitted model. We use the R package rjags to perform this inference. More details about this statistical model are given in S2 File.

While the renewal method better represents the process that generates observed aggregate case reports, it can be computationally intensive. Thus, we also provide a faster, alternative method using simple linear interpolation, described fully in S3 File.

## Daily incidence to $\mathcal{R}_t$

Once daily incidence has been estimated from either data stream, we feed this time series into the function estimate_r() of the package EpiEstim, along with a specific intrinsic general interval distribution. We use the mean value, as well as the 2.5% and 97.5% quantiles, as reported by EpiEstim::estimate_r() as a single estimate of $\mathcal{R}_t$. (Resampling to produce an ensemble $\mathcal{R}_t$ estimate is discussed in the next section.)

Underpinning the EpiEstim::estimate_r() estimation of $\mathcal{R}_t$ is the following equation governing how incidence at the current time, $i(t)$, is modelled by $\mathcal{R}_t$, the generation interval distribution $g$, and past incidence:

$$i(t) = \mathcal{R}_t \sum_{k \geq 1} g(k)i(t-k) \tag{4}$$

Here, $k = 1, \ldots$ is a discrete-time index: incidence is being observed (inferred from reports) at discrete times, $i(t-k)$, and it is being weighted by a discrete generation interval distribution $g(k)$ and scaled by $\mathcal{R}_t$ to calculate current incidence $i(t)$. In other words, current incidence is a function of past incidence (and the generation interval distribution).

The discrete timescale used here is not prescribed (*i.e.* doesn't necessarily have to be daily, weekly, etc.), but Eq 4 shows that the timescales of the generation interval and the observed incidence must match. Many infectious diseases, like influenza and COVID-19, produce generation intervals that are mostly less than a week, and so representing their generation interval distributions on the timescale of weeks (*e.g.* to match weekly reported incidence data input into EpiEstim::estimate_r()) would not yield useful results.

To understand precisely why a coarse generation interval may not yield useful results, let's consider the example of influenza A/H1N1, which has a generation interval distribution

smaller than 7 days in most settings [25] and assume we work with data reported weekly (so the unit of index $k$ is week). In this case, we would need to define the generation interval distribution on a weekly scale as $g(1) = 1$ and $g(k) = 0$ for all $k > 1$ (the generation interval is 0 for any time larger than a week), and so

$$i(t) = \mathcal{R}_t\, i(t-1) \Rightarrow \mathcal{R}_t = i(t)/i(t-1) \tag{5}$$

The parameter $\mathcal{R}_t$ is often used in public health surveillance to determine whether a disease is spreading or receding in a population by comparing it to the $\mathcal{R}_t = 1$ threshold. The crude approximation in Eq 4 would be >1, indicating the disease is spreading, exactly when $i(t) > i(t-1)$, and receding when $i(t) < i(t-1)$. If there is any noise in the incidence time series (inferred from observed reports), which there always is in real data, the approximation in Eq 5 would not be able to distinguish a true increase (or decrease) signal from noise.

For $\mathcal{R}_t$ to be a useful surveillance metric for infectious diseases, the generation interval must be represented in a timescale that describes finely enough the temporal variation of disease transmission. Many infectious disease data (especially respiratory ones) are reported on a coarser timescale (*e.g*, weeks), which is why we have built methods into ern to disaggregate input clinical data (as discussed in section "Estimating daily incidence with clinical data").

### Generating an $\mathcal{R}_t$ ensemble reflecting uncertainty

The package ern accounts for various sources of uncertainty in estimating $\mathcal{R}_t$. There is uncertainty in some inputs used to estimate daily incidence for each data stream, as well as statistical uncertainty incorporated in the daily incidence to $\mathcal{R}_t$ estimate. The latter case is handled by EpiEstim through its Poisson-based model of the renewal equation [26]. The former case is handled by ern. Indeed, ern performs the $\mathcal{R}_t$ calculation repeatedly and then summarizes the results in an ensemble. Each realization of the ensemble involves (re)sampling each uncertain input.

For the wastewater data, the uncertain inputs can be:

- the fecal shedding distribution,

- the intrinsic generation interval distribution.

For the clinical data, the uncertain inputs can be:

- the inferred daily reports,

- the underreporting fraction,

- the incubation period distribution,

- the reporting delay distribution,

- the intrinsic generation interval distribution.

Uncertain distributions are specified for ern as a *family of distributions*, where each distribution parameter has an associated standard deviation. Supported families of distributions include Gamma, Normal, and Log-Normal. One can also specify a Uniform distribution (*e.g.,* for the underreporting proportion). Distribution parameters are assumed to be Gamma-distributed to ensure sampled values (which specify a sampled distribution) are strictly positive. Inferred daily reports are drawn from posterior samples produced by the MCMC fit (if estimated). We sample 300 posterior replicates (using EpiEstim::sample_posterior_R

()) from every single estimate of $\mathcal{R}_t$ (*i.e.*, each realization of the final $\mathcal{R}_t$ ensemble) and calculate by date the mean of those posteriors along with 2.5% and 97.5% quantiles for $\mathcal{R}_t$ to produce a single ensemble time series.

## Results

The package `ern` has two functions with which to estimate the daily effective reproduction number, $\mathcal{R}_t$, for each supported data stream:

- `estimate_R_ww`, which uses the concentration of a pathogen in wastewater over time as the input signal;

- `estimate_R_cl`, which uses the count of clinically reported cases over time as the input signal.

We give an illustration of each method below.

### Example with wastewater data

The function `estimate_R_ww` estimates $\mathcal{R}_t$ from the pathogen concentration measured in wastewater. Its first input, `ww.conc`, is a dataframe with columns `date` (measurement date) and `value` (concentration value) that specifies the pathogen concentration in wastewater over time. The other inputs `dist.fec` and `dist.gi` specify parameters for two families of distributions: one for the fecal shedding rate distribution and the other for the intrinsic generation interval distribution, respectively.

We start by loading a subset of wastewater data that is attached in the `ern` package. This dataset contains daily average concentration data of SARS-CoV-2 (N2 gene), measured in gene copies per milliliter of wastewater, from the Iona Island wastewater treatment plant in Vancouver, British Columbia collected between 7 July 2023 and 5 November 2023 [38]. Note that the type of normalization of the wastewater data (*e.g.,* viral concentration normalized by flow, other biomarkers, suspended solids mass, etc) is left to the user as this choice depends on each sampling site and laboratory methods.

```
R> ww.conc = ern::ww.data
```

This data is plotted in the top panel of Fig 3.

As this example uses the SARS-CoV-2 virus, we can define fecal shedding and generation interval as the following:

```
R> dist.fec = ern::def_dist(
+ dist     = "gamma",
+ mean     = 12.9,
+ mean_sd  = 1.1,
+ shape    = 1.7,
+ shape_sd = 0.27,
+ max      = 33
+ )

R> dist.gi = ern::def_dist(
+ dist     = "gamma",
+ mean     = 6.8,
+ mean_sd  = 0.74,
+ shape    = 2.4,
```

```
+ shape_sd = 0.36,
+ max      = 15
+ )
```

Each distribution family is defined by a structured list:

```
> print(dist.fec)
$dist
[1] "gamma"

$mean
[1] 12.9

$mean_sd
[1] 1.1

$shape
[1] 1.7

$shape_sd
[1] 0.27

$max
[1] 33
```

The first element of each distribution family list, `dist`, gives the shape of the distribution family. The nomenclature of distribution names follows the one used in R (*e.g.,* `gamma` from the R functions `d/r/q/pgamma`). The next four elements give parameters for this family of distributions, stated in terms of the mean and standard deviation, along with an associated standard deviation (`_sd`) for each distribution parameter. The final element of this list, `max` gives the maximum value to be drawn from this distribution; this is where the density is truncated (and then re-normalized to ensure it still sums to 1). This structure for the distribution list applies to Gamma, Normal, and Log-Normal families. For Uniform, `ern` currently supports only the specification of a single distribution (as opposed to a family). In this case, the distribution list specifying a Uniform would have three entires: `dist`, which would be equal to "unif", and then `min` and `max`, to specify the minimum and maximum values with non-zero density (*i.e.,* the support of the Uniform distribution).

We can visualize distributions by calling the function `plot_dist`. This convenience function will plot the mean distribution of the given family, that is, the distribution corresponding to the mean of each distribution parameter in the family. For example, `plot_dist(dist.fec)` was used to produce Fig 2 from the parameters for SARS-CoV-2 specified above.

The function `estimate_R_ww` also takes a number of parameters that give the user control over various components of the $\mathcal{R}_t$ estimation:

- `scaling.factor` is the average number of infections attributable to a unit of pathogen concentration per day. This quantity is typically estimated from i) clinical cases, ii) wastewater concentrations and iii) an "ascertainment rate" that estimates the number of infections missed by clinical surveillance (for example, using serological data).

- `prm.smooth` defines the smoothing settings for the input wastewater data;

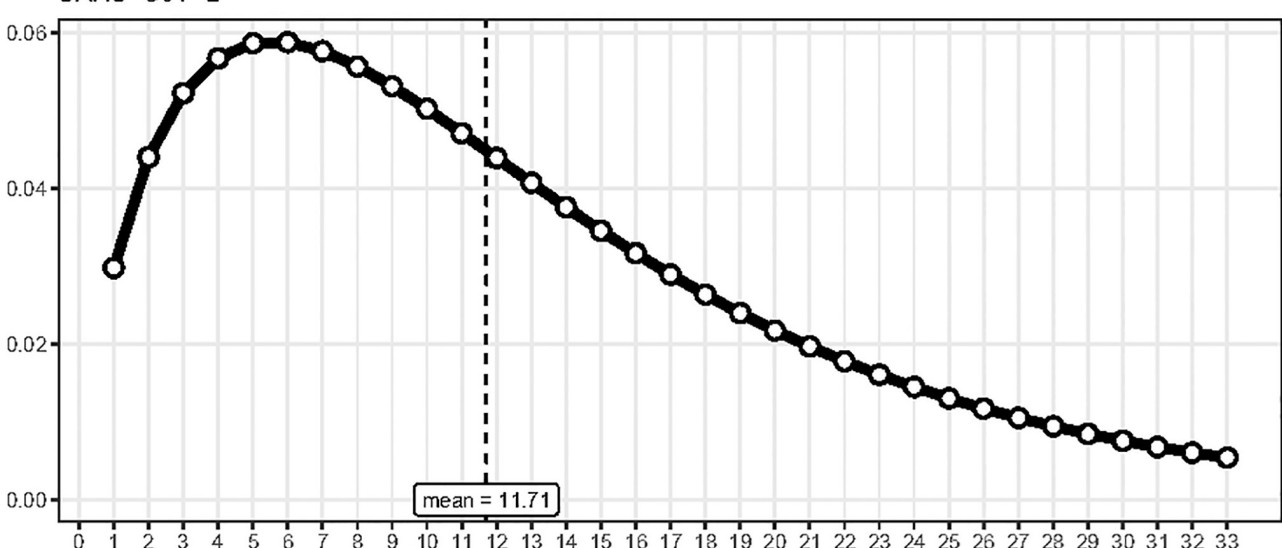

**Fig 2. Fecal shedding distribution example.** A possible choice for the mean fecal shedding distribution used for SARS-CoV-2 wastewater data.

- `prm.R` defines the settings for the $\mathcal{R}_t$ estimates.

```
# Initializing scaling factor
R> scaling.factor = 1

# Initializing smoothing parameters
R> prm.smooth = list(
+ method = 'loess',  # smoothing method
+ align  = 'center', # smoothing alignment
+ span   = 0.30,     # smoothing span (used for loess smoothing only)
+ floor  = 5         # minimum smoothed concentration value
+                    # (optional, LOESS smoothing only)
+ )

# Initializing Rt settings
R> prm.R = list(
+ iter   = 20,             # number of resampling iterations
+                          # to evaluate Rt ensemble
+ CI     = 0.95,           # confidence interval
+ window = 10,             # backward time window for Rt calculations
+ config.EpiEstim = NULL # optional EpiEstim configuration
+                          # for Rt calculations
+ )
```

Once we have specified all of these settings, we can feed them in, along with the input wastewater concentration data and the relevant distributions, to estimate $\mathcal{R}_t$:

```
R> r.estim = estimate_R_ww(
+ ww.conc        = ww.conc,
+ dist.fec       = dist.fec,
+ dist.gi        = dist.gi,
+ scaling.factor = scaling.factor,
+ prm.smooth     = prm.smooth,
+ prm.R          = prm.R
+ )
```

`estimate_R_ww` returns a list with four elements:

- `ww.conc`: the original input of pathogen concentration in wastewater over time

- `ww.smooth`: the smoothed wastewater concentration over time; includes columns:

  - `t`: internal time index

  - `obs`: smoothed value of the observation

  - `date`

- `inc`: the daily incidence inferred over time; includes columns:

  - `date`

  - `mean`: mean of the inferred daily incidence

  - `lwr`, `upr`: lower and upper bounds of the 95% confidence interval for the inferred daily incidence

- `R`: the estimated daily reproduction number over time; includes columns:

  - `date`

  - `mean`: mean $\mathcal{R}_t$ value

  - `lwr`, `upr`: lower and upper bounds of the confidence interval (width as specified in `prm.R`) for $\mathcal{R}_t$

The function `plot_diagnostic_ww` conveniently displays all of the output data to help assess the quality of the $\mathcal{R}_t$ estimates (Fig 3).

## Example with clinical data

As shown in Fig 1, a key feature implemented in `ern` is the ability to handle clinical data that is reported on a time scale that is coarser than the typical generation interval timescale when estimating $\mathcal{R}_t$.

The function `estimate_R_cl` requires a data frame, `cl.data`, with one column for the report date (`date`) and another for the count of clinical reports (`value`). In addition, the user must specify a reporting fraction distribution (`dist.repfrac`) and three distribution families:

- `dist.repdelay`: reporting delay;

- `dist.incub`: incubation period;

- `dist.gi`: intrinsic generation interval.

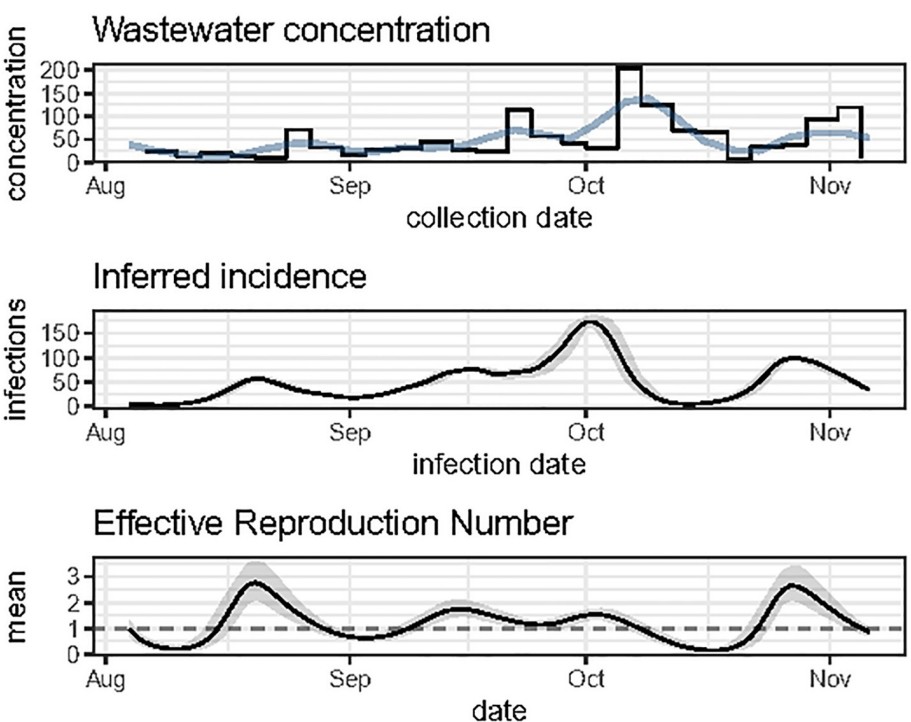

**Fig 3. Output of the function `plot_diagnostic_ww`.** The top panel shows the wastewater concentration data used as input (step line) along with the smoothed version of this time series (curve). The middle panel represents the daily incidence inferred from the smoothed wastewater concentration data (using the Richardson-Lucy deconvolution algorithm). The grey band gives a confidence band reflecting the uncertainty associated with the fecal shedding distribution. (The confidence width is set with `prm.R$CI`.) The estimated incidence is proportional to the parameter `scaling.factor`, here assumed equal to 1. The bottom panel shows the mean $\mathcal{R}_t$ estimates (solid line), along with a 95% confidence interval (grey band) reflecting various sources of uncertainty. The horizontal dashed line represents the $\mathcal{R}_t$ threshold value of 1, which is epidemiologically important.

If input clinical reports are not reported daily, an additional parameter must be provided: `popsize`, representing the size of the population being considered, in order for daily reports to be inferred using the "renewal" method (see S1 File).

A sample of Canadian COVID-19 clinical reports are included in `ern`. This data set includes weekly reports from the provinces of British Columbia, Alberta, Saskatchewan, Manitoba, Ontario, and Quebec, between 1 Feb 2020 and 1 Apr 2023 [39]. As an example, we start by loading a subset of the weekly clinical report data for Quebec:

```
R> # --- data
+ dat <- (ern::cl.data
+   |> dplyr::filter(
+       pt == "qc",
+       dplyr::between(date,
+                   as.Date("2021-06-01"),
+                   as.Date("2021-09-01"))
+   )
)
```

We define distributions for the reporting fraction, reporting delay, incubation period, and intrinsic generation interval:

```
R> # --- distributions
+ # reporting fraction
+ dist.repfrac = ern::def_dist(
+     dist = "unif",
+     min  = 0.1,
+     max  = 0.3
+ )
+ # reporting delay
+ dist.repdelay = ern::def_dist(
+     dist    = 'gamma',
+     mean    = 5,
+     mean_sd = 1,
+     sd      = 1,
+     sd_sd   = 0.1,
+     max     = 10
+ )
+ # incubation period
+ dist.incub = ern::def_dist(
+     dist     = "gamma",
+     mean     = 3.49,
+     mean_sd  = 0.1477,
+     shape    = 8.5,
+     shape_sd = 1.8945,
+     max      = 8
+ )
+ # generation interval
+ dist.gi = ern::def_dist(
+     dist     = "gamma",
+     mean     = 6,
+     mean_sd  = 0.75,
+     shape    = 2.4,
+     shape_sd = 0.3,
+     max      = 10
+ )
```

The data set we are working with reports COVID-19 on a weekly basis, which is substantially longer than the typical generation interval of about 5 days for SARS-CoV-2 [40]. `ern` will estimate daily incidence from non-daily data. We specify the settings for this inference via `prm.daily`:

```
R> # --- settings
+ # daily report inference
+ prm.daily <- list(
+     method = "renewal",
+     popsize = 8.5e6, # Q3 (July 1) 2022 estimate for Quebec
+     burn    = 500,   # "burn-in" for MCMC
+     iter    = 500,   # MCMC iterations after burn-in
```

```
+       chains  = 2,      # number of chains
+       # priors for the R0 distribution (Gamma)
+       prior_R0_shape = 1.1, prior_R0_rate = 0.6,
+       # priors for the alpha distribution
+       prior_alpha_shape = 1, prior_alpha_rate = 1
+ )
```

The `method = "renewal"` setting specifies the use of the renewal-equation-based epidemic model fitted with an MCMC algorithm, described fully in S1 File. This algorithm requires the specification of a total population size, which we source from Statistics Canada for this example [41]. The rest of the arguments in `prm.daily` give settings for the MCMC algorithm. The output of `estimate_R_cl()` has an element called `diagnostic.mcmc` which contains objects that help assess the convergence of the MCMC algorithm. In particular, a warning message is displayed if the Gelman-Rubin statistics [42] of the latent daily incidence variable is above 1.025, prompting the user to increase the number of MCMC iterations.

After the inference of the daily reports is performed, a check is run to ensure that the *posterior* aggregated daily reports are not too different from the *observed* aggregated reports (given as input). The parameter `agg.reldiff.tol` is the maximum tolerance (as a percentage) accepted for the relative difference between the observed and posterior aggregates:

```
R> # daily report inference check
+ prm.daily.check <- list(
+       agg.reldiff.tol = 10
+ )
```

The Bayesian model tends to be most error-prone at the start of the input time series, so after performing this check, `ern` will drop any inferred values before the differences first fall below the specified tolerance. It will not filter out observations after that point to ensure the inferred time series remains daily. It will also produce a warning to ensure the user is aware how many observations were dropped, along with some advice on how to increase the accuracy of the MCMC fit to decrease the number of dropped observations.

Choosing a number of MCMC iterations that is not very large (to avoid long computation times, for example) may lead to daily report posteriors that are not very smooth. This, in turn, can affect the quality of $\mathcal{R}_t$ estimates. Hence, `ern` provides a smoothing of the posterior daily reports in order to improve the quality of $\mathcal{R}_t$ inference. The smoothing parameters are defined as follows:

```
R> # smoothing
+ prm.smooth <- list(
+       method = "rollmean",
+       align  = "center",
+       window = 7
+ )
```

In the example above, the smoothing performs a centered moving average with a sliding window of 7 days. The same smoothing options are available across the wastewater and clinical methods.

We specify the parameters for the $\mathcal{R}_t$ ensemble, just as we did in the wastewater example:

```
R> # Rt computation
+ prm.R <- list(
+   iter          = 20,
+   CI            = 0.95,
+   window        = 7,
+   config.EpiEstim = NULL
+ )
```

Finally, we can call the main `ern` function to estimate $\mathcal{R}_t$ from clinical data:

```
R> r.estim <- estimate_R_cl(
+    dat = cl.data,
+    dist.repdelay  = dist.repdelay,
+    dist.repfrac   = dist.repfrac,
+    dist.incub     = dist.incub,
+    dist.gi        = dist.gi,
+    prm.daily      = prm.daily,
+    prm.daily.check = prm.daily.check,
+    prm.smooth     = prm.smooth,
+    prm.R          = prm.R
+ )
```

`estimate_R_cl` returns a list with four elements:

- `cl.data`: the original input of clinical disease reports over time, with an added column `t` for an internal time index

- `cl.daily`: reports as input for $\mathcal{R}_t$ calculation (inferred daily counts if original inputs were aggregates, smoothed if specified); includes columns:

  - `id`: identifier for each realization (resampling iteration) of the daily report inference

  - `date`: daily date

  - `value`: inferred daily report count

  - `t`: internal time index

- `inferred.agg`: inferred daily reports re-aggregated on the reporting schedule as input in `cl.data`; includes columns:

  - `date`: report date

  - `obs`: original (aggregated) observations

  - `mean.agg`: mean of the aggregated posterior daily reports

  - `lwr.agg`, `upr.agg`: lower and upper bounds of a 95% confidence interval of the aggregated inferred daily reports

- `R`: the estimated daily reproduction number over time; includes columns:

  - `date`

  - `mean`: mean $\mathcal{R}_t$ value

- `lwr`, `upr`: lower and upper bounds of a confidence interval for each $\mathcal{R}_t$ estimate

- `use`: logical flag, `FALSE` denotes estimated $\mathcal{R}_t$ values that may be particularly unreliable as they fall within the maximum time range of one (truncated) generation interval from the start of the clinical report time series

- `diagnostic.mcmc`: a list with various MCMC diagnostics, including

  - `plot.traces`: trace plots for fitted parameters

  - `plot.gelmanrubin`: plot of the Gelman Rubin statistics for fitted parameters

  - `jags.obj`: the JAGS output `mcmc.list` object, as produced by `rjags` [43]

The function `plot_diagnostic_cl` summarises this output (Fig 4).

## Sensitivity analysis for wastewater $\mathcal{R}_t$

We perform a sensitivity analysis of $\mathcal{R}_t$ estimations with wastewater input data to investigate various input choices since the methods used for this data stream are still relatively new. The code to replicate these results is provided in S5 File.

The package `ern` currently includes two smoothing methods: rolling mean and LOESS. Using similar smoothness parameters, *i.e.,* a centered rolling mean on a 5-day window and a span parameter of 0.3 for LOESS, Fig S4–1 in S4 File shows that the $\mathcal{R}_t$ estimates are comparable.

Because of the paucity of clinical studies, there is a fair amount of uncertainty regarding the temporal profile of fecal shedding for respiratory infections. Hence, in Fig S4–2 in S5 File, we show how the $\mathcal{R}_t$ estimates can be significantly impacted by assuming differing profiles based Gamma, normal, uniform, and exponential distribution-like shapes for the fecal shedding distribution.

When the prevalence of infections is low in the population of interest, the epidemic "signal", represented by a low count of clinical reports and/or low viral concentration in wastewater, is dominated by noise. In this case, the estimation of $\mathcal{R}_t$ may be challenging. In Figs S4–3 in S4 File, we illustrate this using wastewater data by estimating $\mathcal{R}_t$ on sample data multiplied by a factor of 0.01, 0.1, 1 and 10. The $\mathcal{R}_t$ estimates are similar for multipliers 1 and 10, but very different (and unreliable) when the multiplier is 0.1 or 0.01, confirming the difficulty of estimating $\mathcal{R}_t$ when prevalence is (very) low.

## Computing time benchmarks

Rapid $\mathcal{R}_t$ estimation can be important in some cases, such as during an epidemic being monitored daily in order to follow its evolution closely and assess the success of ongoing interventions meant to reduce transmission. Here, a computation time of less than a day is key. Rapid $\mathcal{R}_t$ calculation is also important in cases where there are many input datasets. For instance, if one is calculating $\mathcal{R}_t$ with wastewater data across an entire country, they may wish to do so by computing one $\mathcal{R}_t$ per wastewater sampling location (it can be difficult to meaningfully combine wastewater data sampled from different sites into a single signal). Here, it is important for the $\mathcal{R}_t$ calculation to be quick so that one can produce $\mathcal{R}_t$ estimates for a large number of wastewater sampling locations in a reasonable amount of time.

As an example, Table 2 shows computing times to calculate $\mathcal{R}_t$ with different R packages using either weekly clinical case reports or viral concentration in wastewater. These times are simply meant to illustrate the order of magnitudes of the calculation times, and do not represent a thorough benchmarking exercise. In this example, estimates for wastewater-based $\mathcal{R}_t$

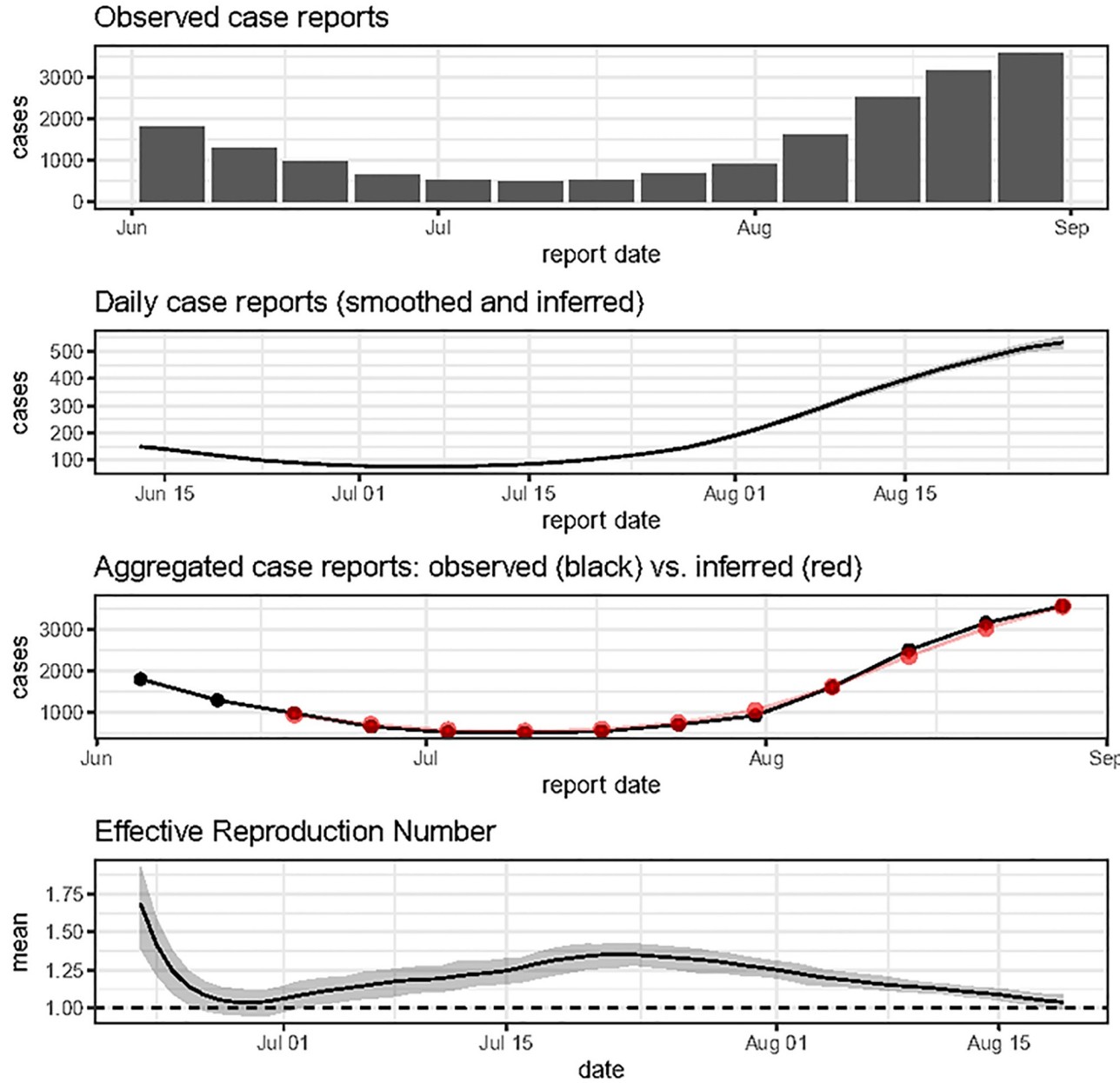

**Fig 4. Output of the function `plot_diagnostic_cl`.** The top panel shows the observed case report data used as input. The second panel from the top shows daily reports, smoothed and, in this case, inferred from the input aggregate (weekly) reports. When this inference is made, this panel also summarises the ensemble of daily report time series with a grey band, whose limits are given by the 2.5% and 97.5% percentiles by day. The second panel from the bottom appears only in the case where the input data is coarser than daily and compares the observed (aggregate) reports (black points) to aggregates from inferred daily reports (red points with 95% confidence bars), so that the user can check whether inferred daily reports are plausible against the input data. The bottom panel shows the mean $\mathcal{R}_t$ estimates (solid line), along with a 95% confidence interval (grey band) reflecting various sources of uncertainty. The horizontal dashed line represents the $\mathcal{R}_t$ threshold value of 1, which is epidemiologically important.

take about one second with `ern` compared to about four minutes with `EpiSewer`. The latter uses a Hamiltonian Markov Chain Monte Carlo (via Stan) to estimate latent variables, which is much more computationally intensive than the simple deconvolution performed in `ern`. For $\mathcal{R}_t$ estimation based on weekly clinical reports, the computing time is on the order of one

**Table 2. Sample computing times (in seconds) for $R_t$ estimates using different R packages.** The wastewater data is taken from the data set shipped with the package, which consists of four months of daily SARS-CoV-2 concentration measurements for the city of Zurich. The clinical data are simulated weekly reports. See S6 File for more details.

| | R package | data type | method (daily report inference) | compute time (s) |
|---|---|---|---|---|
| 1 | ern | wastewater | - | 0.85 |
| 2 | EpiSewer | wastewater | - | 251.23 |
| 3 | ern | clinical | linear | 1.89 |
| 4 | ern | clinical | renewal | 20.72 |
| 5 | EpiEstim | clinical | expectation maximization | 0.81 |

second for both ern (using the linear method) and EpiEstim. The code to reproduce this example is given in S6 File.

## Discussion and conclusions

The R package ern was designed with public health practitioners in mind, specifically to provide them with a tool to estimate, in a user-friendly way, the effective reproduction number $\mathcal{R}_t$ from typical clinical reports and/or data reporting pathogen concentration in wastewater. The inferences for $\mathcal{R}_t$ rely on various distributions (*e.g.,* fecal shedding, incubation period, generation interval) that are rarely perfectly known. To reflect this uncertainty, these distributions are defined as *family* of distributions and the estimation process samples from those families to propagate this source of uncertainty into the final $\mathcal{R}_t$ estimates. Clinical cases of infectious diseases are rarely reported on a daily basis despite being the most natural time unit (at least for respiratory diseases) in $\mathcal{R}_t$ models. The package ern accepts non-daily clinical reports and can infer daily incidence using a genuine transmission model.

The methods implemented in ern to estimate $\mathcal{R}_t$ from clinical or wastewater data are similar to other existing methods. For example, the deconvolution of the incubation period and reporting delays in ern use the same Richardson-Lucy algorithm as in [15, 16]. The LOESS or rolling mean smoothing of the wastewater data as a way to preprocess the data to reduce the noise is also use broadly. Indeed, the R package ern leverages previous works and focuses its scientific contribution on bringing these different methodological approaches into a single, consistent, user-friendly package.

There are several limitations of the ern package. For clinical inputs, the renewal method depends on JAGS, which may not be straightforward to install for the average user. The computing time when using aggregated clinical reports and the renewal method may be too long for some applications. Moreover, the renewal method does not have a time-dependent transmission parameter in its current implementation, so estimating $\mathcal{R}_t$ using this method is appropriate for a single epidemic wave without any significant change in transmission (for example, a typical seasonal influenza wave in a non-tropical region). The linear method can handle temporal changes in transmission, though it may not always infer a realistic epidemic curve for inferred daily reports.

Another limitation is that the model in ern does not have the latent incidence as a random variable when estimating $\mathcal{R}_t$ from wastewater data (unlike, for example, the R package EpiSewer), so this uncertainty is not accounted for. Even if the uncertainty of the fecal shedding distribution is propagated, it does not capture the full scale of uncertainty. This can be problematic for real-time surveillance because the uncertainty for $\mathcal{R}_t$ estimates may be underestimated for dates close to the estimation time.

For wastewater inputs, the scaling factor used to convert between prevalence and viral concentration in wastewater is difficult to estimate in practice. Ideally, one would need, over multiple days, i) an accurate estimate of the actual prevalence in the catchment area from (extensive) clinical surveillance and ii) viral concentration measurements over the same period. The scaling factor would then be proportional to the ratio of prevalence over concentration (and depending on the laboratory method used to measure the viral concentration, additional normalization, for instance by flow rate or suspended solid mass, may be required).

ern currently allows users to define a particular distribution for fecal shedding kinetics. Studies examining SARS-CoV-2 shedding have shown that fecal shedding kinetics can vary among infected individuals [44, 45]. Moreover, the scaling factor in ern is held constant over time, which may not be realistic as new viral lineages emerge and the immune profile of the population evolves over time; both of these factors can affect pathogen shedding in wastewater. As a result, the "inferred incidence" estimated by ern (using the output estimate_R_ww (...) $inc) must be interpreted carefully.

Wastewater sample concentration can also be affected by environmental and structural factors of sewer systems. Flow from rainfall and snowmelt can dilute sample concentration readings [46] and sewer transit time can impact the rate at which viral particles degrade prior to sample collection [47].

Future versions of ern will attempt to address the above limitations.

In conclusion, the R package ern aims to provide a relatively user-friendly environment to empower public health professionals with a tool to estimate the effective reproduction number $\mathcal{R}_t$ from clinical and wastewater-based data.

## Computational details

The results in this paper were obtained using R version 4.3.1 with packages EpiEstim version 2.4, rjags version 4–14, and the software JAGS version 4.3.1. R itself and all packages used are available from the Comprehensive R Archive Network (CRAN) at https://CRAN.R-project.org/.

## Supporting information

**S1 File. Methodological differences when inferring daily incidence.**
(PDF)

**S2 File. Bayesian model to infer daily clinical report count.**
(PDF)

**S3 File. Linear interpolation to infer daily clinical report count.**
(PDF)

**S4 File. Sensitivity analysis to selected parameters.**
(PDF)

**S5 File. R code to perform sensitivity analyses presented in S4 File.**
(R)

**S6 File. R code to evaluate the computing time of selected R packages that estimate the effective reproduction number.**
(R)

**S7 File. R code to associated with the methodological differences presented in S1 File.**
(R)

## Acknowledgments

We thank Shokoofeh Nourbakhsh for testing early versions of the package. We also thank the reviewers for their insightful comments that sparked improvements in this article and in the ern package presented here.

## Author Contributions

**Conceptualization:** David Champredon.

**Formal analysis:** Irena Papst, Warsame Yusuf.

**Methodology:** David Champredon, Irena Papst, Warsame Yusuf.

**Resources:** David Champredon.

**Software:** David Champredon, Irena Papst, Warsame Yusuf.

**Supervision:** David Champredon.

**Validation:** David Champredon.

**Writing – original draft:** David Champredon.

**Writing – review & editing:** David Champredon, Irena Papst, Warsame Yusuf.

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
