## [Decision Letter · Decision Letter 0]

31 Jan 2024

PONE-D-24-01907ern: an R package to estimate the effective reproduction number using clinical and wastewater surveillance dataPLOS ONE

Dear Dr. Champredon,

Thank you for submitting your manuscript to PLOS ONE. After careful consideration, we feel that it has merit but does not fully meet PLOS ONE’s publication criteria as it currently stands. Therefore, we invite you to submit a revised version of the manuscript that addresses the points raised during the review process.

We look forward to receiving your revised manuscript.

Kind regards,

Salim Heddam

Academic Editor

PLOS ONE

Journal Requirements:

2. We are unable to open your Supporting Information file [suppl_S1_interpolation_impact.R]. Please kindly revise as necessary and re-upload.

**Additional Editor Comments:**

Reviewer 1#:

please abstract must be more clear

material and methods not Models and software 93

discussion must be wrirten more advanced refernce with more knowedge about your artcle

conculsion must be written with detail

Reviewer 2#:

The authors present the R package ern for the estimation of the effective reproduction number from wastewater or aggregated clinical surveillance data.

The package provides a framework for an efficient and quicker estimation of effective reproduction number using a user-friendly interface.

The manuscript is well-written and makes a relevant contribution to the field.

I thoroughly enjoyed reviewing this manuscript and only have some minor requests for revision, as follows:

Lines 39 to 42: Do not start a sentence using a citation number. In Line 39, you may write, "Huisman et al. [14] proposed a method....". Do the same for lines 40, 41 and 42.

Line 261: "...for the for the ...". Delete the repetition.

Reviewer 3#:

1- It would have been better to talk about the Rt factor, which was mentioned in the research, in numerical terms, with something simple in the abstract

2-It was possible to dispense with some paragraphs in figure or table in the introduction

3-The researcher did not provide a research review of references that address the same topic, even in a simple way

4- It is possible to clarify the work algorithm in the form of clear points or in an algorithmic form, on the basis of which the steps of the example are clarified

5- The discussion was narrative and did not clarify any future idea or plan of action for researchers working in the future in the same field and what difficulties they may face.

Reviewer 4#:

1. Describe dataset features in more details and its total size and size of (train/test) as a table.

2. Pseudocode / Flowchart and algorithm steps need to be inserted.

3. Time spent need to be measured in the experimental results.

4. Limitation and Discussion Sections need to be inserted.

5. The parameters used for the analysis must be provided in table

6. The architecture of the proposed model must be provided

7. Address the accuracy/improvement percentages in the abstract and in the conclusion sections, as well as the significance of these results.

8. The authors need to make a clear proofread to avoid grammatical mistakes and typo errors.

9. Add future work in last section (conclusion) (if any)

10. To improve the Related Work and Introduction sections authors are recommended to review this highly related research work paper:

a) Optimizing epileptic seizure recognition performance with feature scaling and dropout layers

b) Optimizing classification of diseases through language model analysis of symptoms

c) Predicting female pelvic tilt and lumbar angle using machine learning in case of urinary incontinence and sexual dysfunction

d) Utilizing convolutional neural networks to classify monkeypox skin lesions

e) Hepatitis C Virus prediction based on machine learning framework: a real-world case study in Egypt

Reviewer 5#:

I would like to thank the authors for putting together the R package ern and for submitting this accompanying manuscript. I agree with the assessment that there is a need for user-friendly statistical software to estimate reproduction numbers from wastewater concentration measurements. This manuscript explains the motivation for developing the package ern, i.e. to provide a dedicated interface to estimate Rt from wastewater and clinical case data. It then describes the statistical approach used and presents a vignette-style illustration of the main functionalities of the package. The authors also propose a new approach to disaggregate non-daily case counts for subsequent Rt estimation.

I have read the manuscript in detail and I have tested the package both using the example data from Canada as provided by the authors, and using wastewater data from Switzerland. I tried to structure my review into manuscript-related and package/method-related major comments, plus a collection of minor comments.

For transparency, I am a (co)author of two packages mentioned in my review, i.e. the package "estimateR" and the package "EpiSewer".

Kind regards

Adrian Lison

## Major points manuscript

First let me say that I found the manuscript clean and well-written.

### Related work

I think the manuscript can provide more details on what is methodologically novel and what not. You mention that the method implemented is similar to the method by Huisman et al., but aside from the approach to disaggregate non-daily case data, it seems at first glance to be EXACTLY the method by Huisman et al., i.e. as detailed in https://doi.org/10.7554/eLife.71345 for case data and in https://doi.org/10.1289/EHP10050 for wastewater data (LOESS smoothing, deconvolution using Richardson-Lucy algorithm, scaling, R estimation using EpiEstim, uncertainty quantification via resampling). I am raising this not only because of attribution but because it is important to clearly describe the differences and similarities between related methods such that readers can compare them properly.

Furthermore, there are several related works that are worth mentioning in my opinion.

First, the method by Huisman et al. has also been implemented as an open source R package (https://doi.org/10.1186/s12859-023-05428-4) "estimateR", and can similarly be used to estimate Rt from wastewater and case data (see e.g. https://ibz-shiny.ethz.ch/wastewaterRe/). Compared to the package ern, the interface and plotting functionality of estimateR are not explicitly tailored to wastewater data, therefore I think that ern is more user-friendly for this domain. Also, estimateR offers no option to disaggregate non-daily case counts. Aside from that, I think that estimateR and ern are highly similar since they are based on almost the same method.

Aside from ern, there are also other R packages for modeling wastewater data, including the package EpiSewer (https://zenodo.org/doi/10.5281/zenodo.10569101), your own package wem (https://github.com/phac-nml-phrsd/wem/tree/main), and the Covid19 Wastewater Analysis Package (https://github.com/UW-Madison-DSI/Covid19Wastewater/tree/main), although the latter does not produce Rt estimates. I do not think a detailed comparison or benchmarking of these packages is necessary, but a short discussion of their differences would be useful. I believe it is valuable to give potential users an overview over available options.

### Sensitivity analyses

I liked the illustrations of the package, but what I missed are some sensitivity analyses that give users an idea of what behavior to expect in different situations and what limitations the method might have. Some interesting analyses that I can think of would for example be about the smoothing (what role do the hyperparameters play, how do moving average and LOESS compare, how does the interpolation deal with larger gaps of data), or what happens if the fecal shedding distribution is misspecified, or how the method performs when concentrations are low. I know that such kinds of analyses require some work, but I think having some sensitivity analyses would add much value to the paper. You can then also point to these analyses from the package documentation.

### Limitations

Lines 22f: I agree that wastewater has several advantages over clinical data and does not have the same biases, but it would be good to shortly mention also potential biases of wastewater. In particular I am worried that the statement "Fecal shedding occurs passively and irrespective of the symptomatic status of the infected individual" could be misunderstood by readers. While it is true that asymptomatic patients also seem to shed into wastewater, there is also large variation in shedding loads and distributions between patients and it is not yet clear what the main factors are (see e.g. https://doi.org/10.1016/j.scitotenv.2020.141364 and https://doi.org/10.1128/msphere.00132-23). Wastewater concentrations could also be to a large part be driven by "supershedders" and we don't know how representative this subgroup is of the overall population. Other sources of bias worth mentioning are changing populations in the catchments and environmental factors like rainfall.

Scaling factor: Can you give some more details on how you would choose this in practice, and what the implications of a potential misspecification are (for example that a constant factor of misspecification will strongly bias incidence estimates but at least not bias Rt except in certain edge cases).

For the above reason, I would stress in the manuscript (and also in the package documentation) that the inferred incidence directly depends on the hard-to-estimate scaling factor and must therefore be carefully interpreted. Otherwise there is a risk that people will use this to estimate prevalence etc.

I suggest to add unit information to the concentration and scaling factor. For example, what are the typical units of the exemplary data in ww.input?

### Disaggregation

The proposed method for disaggregation of non-daily cases and the comparison with the method by Nash et al. is quite interesting. Based on your illustration, it seems like the method by Nash et al. could have important limitations not found in the original study by Nash et al.. Since this would be a strong result, can you provide

more details, e.g. which sliding window was used in the example? Also in Figure 5, can you also plot Rt estimated from the daily incidence time series?

In your model for inferring daily case counts, you do not seem to account for potential changes in transmission other than due to susceptible depletion. What happens if you fit this over longer time periods with multiple waves or time series with strong non-pharmaceutical interventions?

Lines 379f: I think this is a rather problematic approach - drawing not enough posterior samples and then applying smoothing to improve the irregular posterior. I think this can easily lead to an unrepresentative posterior and also distort the uncertainty estimates.

In the introduction, you mention long runtimes of epidemia and EpiNow2 as disadvantages to overcome, but the disaggregation of ern also requires MCMC sampling. How do the runtimes compare on non-daily data, is ern still considerably faster?

## Major points package / method

The package ern currently only seems to offer a fixed scaling factor. Explicitly supporting a time-varying scaling factor to account for flow would be great. Scaling concentrations by daily flow volumes at the treatment plant can be quite important in my experience because there can be a strong effect of dilution of the viral particles by rainfall etc. on the measured concentration.

Lines 173: What output of EpiEstim is used for Rt? Do you use the estimated mean of Rt? Or do you draw samples from the posterior Gamma distribution estimated by EpiEstim?

I noticed that the Rt estimates provided by ern do not have higher uncertainty towards the present, although this should definitely be the case (Rt of today cannot be estimated with the same accuracy as Rt of last week because of delays in fecal shedding / reporting). Do you account for uncertainty in the deconvolution step?

I think it is a great feature that ern also supports uncertain distributions. One question I had is if you could also support correlated parameters. At the moment, the mean and sd of a distribution get drawn independently, but they may be correlated in reality. Another scenario that may be quite realistic is that users have several different distributions / parameters from the literature. In this case you could allow users to provide a list of distributions from which to draw with replacement. These are just suggestions, not requests for this paper.

I am still a bit unsure about the default distributions provided in the package. On the one hand, this is a practical feature, but on the other hand: how are you planning to maintain/update this epidemiological data? They seem to be hard-coded in the package, so if newer distributions become available or old distributions are invalidated by new research, users will only get the updates if they install a newer version of the package. It would be good to comment on this.

I am particularly skeptical of providing default reporting delay distributions or reporting proportions as they will differ a lot between surveillance systems / countries etc.

I currently find it hard to define custom distributions with my own parameters. Functions like def_dist_incubation_period only accept the name of a pathogen as input. I know I can also construct a list with custom parameters myself, but a constructor function with the relevant parameters would be helpful.

There is an argument for subtypes/variants in def_dist_fecal_shedding, but it does not seem to do anything, I always get back the same distribution.

Also, def_dist_reporting_fraction does not accept a custom value and assumes a uniform distribution between 0.1 and 0.3 per default. This seems quite arbitrary to me.

I like the diagnostic plots, but having the option to produce individual plots for concentration, incidence, Rt etc. would be valuable. They are currently all merged into one plot via the patchwork package, making it hard to customize individual plots.

## Minor points

Line 5: "It differs from the basic reproduction number, R0, in that it takes into account the level of susceptibility in the population at a given point in time." Maybe say more generally that it accounts for "changes in transmission" - this may be due to changing susceptibility or other factors like different contact patterns, infection control measures etc.

Lines 21f: I would also mention digital droplet PCR as an alternative quantification method. Also, maybe shortly mention that viral RNA is first extracted from the wastewater sample using various laboratory methods.

Line 78: EpiNow2 and epidemia do not use the rstan package, they use cmdstanr package. I would just write "stan".

Lines 113: This is an important point and well explained.

Lines 166f: This paragraph felt a bit informal and was difficult to understand. Is your main point that it is important to estimate Rt using the number of cases by time of infection, not by time of report / time of sample?

I personally don't find the name of the function ern::ww.input very intuitive, I would not have expected that this returns example data! Also, can you provide more details in the function documentation, e.g. what the column "pt" means and what the unit of the concentration is in this example?

Lines 137f, reporting delay: I would say more clearly that this is the delay between symptom onset and case report.

As a suggestion, you could register the current version of the package in an online archive like zenodo. This will give you a DOI for the package, which can then be referenced in the manuscript.

## References

1. Huisman, J. S. _et al._ Estimation and worldwide monitoring of the effective reproductive number of SARS-CoV-2. _eLife_ **11**, e71345 (2022).

2. Huisman, J. S. _et al._ Wastewater-Based Estimation of the Effective Reproductive Number of SARS-CoV-2. _Environmental Health Perspectives_ **130**, 057011 (2022).

3. Scire, J. _et al._ estimateR: an R package to estimate and monitor the effective reproductive number. _BMC Bioinformatics_ **24**, 310 (2023).

4. Lison, A. EpiSewer: Estimate Reproduction Numbers from Wastewater Measurements. Zenodo (2024).

5. Jones, D. L. _et al._ Shedding of SARS-CoV-2 in feces and urine and its potential role in person-to-person transmission and the environment-based spread of COVID-19. _Science of The Total Environment_ **749**, 141364 (2020).

6. Arts, P. J. _et al._ Longitudinal and quantitative fecal shedding dynamics of SARS-CoV-2, pepper mild mottle virus, and crAssphage. _mSphere_ **8**, e00132-23 (2023).

7. Nash, R. K., Bhatt, S., Cori, A. & Nouvellet, P. Estimating the epidemic reproduction number from temporally aggregated incidence data: A statistical modelling approach and software tool. _PLOS Computational Biology_ **19**, e1011439 (2023).

Reviewer 6#:

(See attached PDF for a full evaluation.)

The authors presented an R software package, which implements statistical methods to estimate the actual number of new infections using the number of reported cases or the wastewater data. It is important to note that the package allows the input data to be sampled by a period higher then one day (e.g., aggregated weekly data is also acceptable). Still, the output is a daily time series, which allows to estimate the effective reproduction number using the already existing tool, EpiEstim. To estimate the hidden time-series (i.e., the unknown input) from the measured output, the Authors applied a deconvolution using an existing Richardson-Lucy implementation. As far as I know, this technique is equivalent to a dynamic inversion, which was already used to infer the effective reproduction number.

Although I cannot detect any scientific contribution in this manuscript, the ``attached'' R package may be useful for a certain community (e.g., public health practitioners), and the software description in the main body is clear and didactic.

The manuscript has therefore a raison d'être, possibly not in such a high impact journal (but the Editor is the final judge on that).

Anyway, the Authors need to be better justify why their software tool is preferable or more convenient compared to other existing packages.

Reviewers' comments:

Reviewer's Responses to Questions

**Comments to the Author**

1. Is the manuscript technically sound, and do the data support the conclusions?

Reviewer #1: Yes

Reviewer #2: Yes

Reviewer #3: Yes

Reviewer #4: Partly

Reviewer #5: Yes

Reviewer #6: Yes

2. Has the statistical analysis been performed appropriately and rigorously? 

Reviewer #1: Yes

Reviewer #2: Yes

Reviewer #3: Yes

Reviewer #4: Yes

Reviewer #5: Yes

Reviewer #6: Yes

3. Have the authors made all data underlying the findings in their manuscript fully available?

Reviewer #1: No

Reviewer #2: Yes

Reviewer #3: Yes

Reviewer #4: Yes

Reviewer #5: Yes

Reviewer #6: Yes

4. Is the manuscript presented in an intelligible fashion and written in standard English?

Reviewer #1: No

Reviewer #2: Yes

Reviewer #3: Yes

Reviewer #4: Yes

Reviewer #5: Yes

Reviewer #6: Yes

5. Review Comments to the Author

Reviewer #1: please abstract must be more clear

material and methods not Models and software 93

discussion must be wrirten more advanced refernce with more knowedge about your artcle

conculsion must be written with detail

Reviewer #2: The authors present the R package ern for the estimation of the effective reproduction number from wastewater or aggregated clinical surveillance data.

The package provides a framework for an efficient and quicker estimation of effective reproduction number using a user-friendly interface.

The manuscript is well-written and makes a relevant contribution to the field.

I thoroughly enjoyed reviewing this manuscript and only have some minor requests for revision, as follows:

Lines 39 to 42: Do not start a sentence using a citation number. In Line 39, you may write, "Huisman et al. [14] proposed a method....". Do the same for lines 40, 41 and 42.

Line 261: "...for the for the ...". Delete the repetition.

Reviewer #3: 1- It would have been better to talk about the Rt factor, which was mentioned in the research, in numerical terms, with something simple in the abstract

2-It was possible to dispense with some paragraphs in figure or table in the introduction

3-The researcher did not provide a research review of references that address the same topic, even in a simple way

4- It is possible to clarify the work algorithm in the form of clear points or in an algorithmic form, on the basis of which the steps of the example are clarified

5- The discussion was narrative and did not clarify any future idea or plan of action for researchers working in the future in the same field and what difficulties they may face.

Reviewer #4: 1. Describe dataset features in more details and its total size and size of (train/test) as a table.

2. Pseudocode / Flowchart and algorithm steps need to be inserted.

3. Time spent need to be measured in the experimental results.

4. Limitation and Discussion Sections need to be inserted.

5. The parameters used for the analysis must be provided in table

6. The architecture of the proposed model must be provided

7. Address the accuracy/improvement percentages in the abstract and in the conclusion sections, as well as the significance of these results.

8. The authors need to make a clear proofread to avoid grammatical mistakes and typo errors.

9. Add future work in last section (conclusion) (if any)

10. To improve the Related Work and Introduction sections authors are recommended to review this highly related research work paper:

a) Optimizing epileptic seizure recognition performance with feature scaling and dropout layers

b) Optimizing classification of diseases through language model analysis of symptoms

c) Predicting female pelvic tilt and lumbar angle using machine learning in case of urinary incontinence and sexual dysfunction

d) Utilizing convolutional neural networks to classify monkeypox skin lesions

e) Hepatitis C Virus prediction based on machine learning framework: a real-world case study in Egypt

Reviewer #5: I would like to thank the authors for putting together the R package ern and for submitting this accompanying manuscript. I agree with the assessment that there is a need for user-friendly statistical software to estimate reproduction numbers from wastewater concentration measurements. This manuscript explains the motivation for developing the package ern, i.e. to provide a dedicated interface to estimate Rt from wastewater and clinical case data. It then describes the statistical approach used and presents a vignette-style illustration of the main functionalities of the package. The authors also propose a new approach to disaggregate non-daily case counts for subsequent Rt estimation.

I have read the manuscript in detail and I have tested the package both using the example data from Canada as provided by the authors, and using wastewater data from Switzerland. I tried to structure my review into manuscript-related and package/method-related major comments, plus a collection of minor comments.

For transparency, I am a (co)author of two packages mentioned in my review, i.e. the package "estimateR" and the package "EpiSewer".

Kind regards

Adrian Lison

## Major points manuscript

First let me say that I found the manuscript clean and well-written.

### Related work

I think the manuscript can provide more details on what is methodologically novel and what not. You mention that the method implemented is similar to the method by Huisman et al., but aside from the approach to disaggregate non-daily case data, it seems at first glance to be EXACTLY the method by Huisman et al., i.e. as detailed in https://doi.org/10.7554/eLife.71345 for case data and in https://doi.org/10.1289/EHP10050 for wastewater data (LOESS smoothing, deconvolution using Richardson-Lucy algorithm, scaling, R estimation using EpiEstim, uncertainty quantification via resampling). I am raising this not only because of attribution but because it is important to clearly describe the differences and similarities between related methods such that readers can compare them properly.

Furthermore, there are several related works that are worth mentioning in my opinion.

First, the method by Huisman et al. has also been implemented as an open source R package (https://doi.org/10.1186/s12859-023-05428-4) "estimateR", and can similarly be used to estimate Rt from wastewater and case data (see e.g. https://ibz-shiny.ethz.ch/wastewaterRe/). Compared to the package ern, the interface and plotting functionality of estimateR are not explicitly tailored to wastewater data, therefore I think that ern is more user-friendly for this domain. Also, estimateR offers no option to disaggregate non-daily case counts. Aside from that, I think that estimateR and ern are highly similar since they are based on almost the same method.

Aside from ern, there are also other R packages for modeling wastewater data, including the package EpiSewer (https://zenodo.org/doi/10.5281/zenodo.10569101), your own package wem (https://github.com/phac-nml-phrsd/wem/tree/main), and the Covid19 Wastewater Analysis Package (https://github.com/UW-Madison-DSI/Covid19Wastewater/tree/main), although the latter does not produce Rt estimates. I do not think a detailed comparison or benchmarking of these packages is necessary, but a short discussion of their differences would be useful. I believe it is valuable to give potential users an overview over available options.

### Sensitivity analyses

I liked the illustrations of the package, but what I missed are some sensitivity analyses that give users an idea of what behavior to expect in different situations and what limitations the method might have. Some interesting analyses that I can think of would for example be about the smoothing (what role do the hyperparameters play, how do moving average and LOESS compare, how does the interpolation deal with larger gaps of data), or what happens if the fecal shedding distribution is misspecified, or how the method performs when concentrations are low. I know that such kinds of analyses require some work, but I think having some sensitivity analyses would add much value to the paper. You can then also point to these analyses from the package documentation.

### Limitations

Lines 22f: I agree that wastewater has several advantages over clinical data and does not have the same biases, but it would be good to shortly mention also potential biases of wastewater. In particular I am worried that the statement "Fecal shedding occurs passively and irrespective of the symptomatic status of the infected individual" could be misunderstood by readers. While it is true that asymptomatic patients also seem to shed into wastewater, there is also large variation in shedding loads and distributions between patients and it is not yet clear what the main factors are (see e.g. https://doi.org/10.1016/j.scitotenv.2020.141364 and https://doi.org/10.1128/msphere.00132-23). Wastewater concentrations could also be to a large part be driven by "supershedders" and we don't know how representative this subgroup is of the overall population. Other sources of bias worth mentioning are changing populations in the catchments and environmental factors like rainfall.

Scaling factor: Can you give some more details on how you would choose this in practice, and what the implications of a potential misspecification are (for example that a constant factor of misspecification will strongly bias incidence estimates but at least not bias Rt except in certain edge cases).

For the above reason, I would stress in the manuscript (and also in the package documentation) that the inferred incidence directly depends on the hard-to-estimate scaling factor and must therefore be carefully interpreted. Otherwise there is a risk that people will use this to estimate prevalence etc.

I suggest to add unit information to the concentration and scaling factor. For example, what are the typical units of the exemplary data in ww.input?

### Disaggregation

The proposed method for disaggregation of non-daily cases and the comparison with the method by Nash et al. is quite interesting. Based on your illustration, it seems like the method by Nash et al. could have important limitations not found in the original study by Nash et al.. Since this would be a strong result, can you provide

more details, e.g. which sliding window was used in the example? Also in Figure 5, can you also plot Rt estimated from the daily incidence time series?

In your model for inferring daily case counts, you do not seem to account for potential changes in transmission other than due to susceptible depletion. What happens if you fit this over longer time periods with multiple waves or time series with strong non-pharmaceutical interventions?

Lines 379f: I think this is a rather problematic approach - drawing not enough posterior samples and then applying smoothing to improve the irregular posterior. I think this can easily lead to an unrepresentative posterior and also distort the uncertainty estimates.

In the introduction, you mention long runtimes of epidemia and EpiNow2 as disadvantages to overcome, but the disaggregation of ern also requires MCMC sampling. How do the runtimes compare on non-daily data, is ern still considerably faster?

## Major points package / method

The package ern currently only seems to offer a fixed scaling factor. Explicitly supporting a time-varying scaling factor to account for flow would be great. Scaling concentrations by daily flow volumes at the treatment plant can be quite important in my experience because there can be a strong effect of dilution of the viral particles by rainfall etc. on the measured concentration.

Lines 173: What output of EpiEstim is used for Rt? Do you use the estimated mean of Rt? Or do you draw samples from the posterior Gamma distribution estimated by EpiEstim?

I noticed that the Rt estimates provided by ern do not have higher uncertainty towards the present, although this should definitely be the case (Rt of today cannot be estimated with the same accuracy as Rt of last week because of delays in fecal shedding / reporting). Do you account for uncertainty in the deconvolution step?

I think it is a great feature that ern also supports uncertain distributions. One question I had is if you could also support correlated parameters. At the moment, the mean and sd of a distribution get drawn independently, but they may be correlated in reality. Another scenario that may be quite realistic is that users have several different distributions / parameters from the literature. In this case you could allow users to provide a list of distributions from which to draw with replacement. These are just suggestions, not requests for this paper.

I am still a bit unsure about the default distributions provided in the package. On the one hand, this is a practical feature, but on the other hand: how are you planning to maintain/update this epidemiological data? They seem to be hard-coded in the package, so if newer distributions become available or old distributions are invalidated by new research, users will only get the updates if they install a newer version of the package. It would be good to comment on this.

I am particularly skeptical of providing default reporting delay distributions or reporting proportions as they will differ a lot between surveillance systems / countries etc.

I currently find it hard to define custom distributions with my own parameters. Functions like def_dist_incubation_period only accept the name of a pathogen as input. I know I can also construct a list with custom parameters myself, but a constructor function with the relevant parameters would be helpful.

There is an argument for subtypes/variants in def_dist_fecal_shedding, but it does not seem to do anything, I always get back the same distribution.

Also, def_dist_reporting_fraction does not accept a custom value and assumes a uniform distribution between 0.1 and 0.3 per default. This seems quite arbitrary to me.

I like the diagnostic plots, but having the option to produce individual plots for concentration, incidence, Rt etc. would be valuable. They are currently all merged into one plot via the patchwork package, making it hard to customize individual plots.

## Minor points

Line 5: "It differs from the basic reproduction number, R0, in that it takes into account the level of susceptibility in the population at a given point in time." Maybe say more generally that it accounts for "changes in transmission" - this may be due to changing susceptibility or other factors like different contact patterns, infection control measures etc.

Lines 21f: I would also mention digital droplet PCR as an alternative quantification method. Also, maybe shortly mention that viral RNA is first extracted from the wastewater sample using various laboratory methods.

Line 78: EpiNow2 and epidemia do not use the rstan package, they use cmdstanr package. I would just write "stan".

Lines 113: This is an important point and well explained.

Lines 166f: This paragraph felt a bit informal and was difficult to understand. Is your main point that it is important to estimate Rt using the number of cases by time of infection, not by time of report / time of sample?

I personally don't find the name of the function ern::ww.input very intuitive, I would not have expected that this returns example data! Also, can you provide more details in the function documentation, e.g. what the column "pt" means and what the unit of the concentration is in this example?

Lines 137f, reporting delay: I would say more clearly that this is the delay between symptom onset and case report.

As a suggestion, you could register the current version of the package in an online archive like zenodo. This will give you a DOI for the package, which can then be referenced in the manuscript.

## References

1. Huisman, J. S. _et al._ Estimation and worldwide monitoring of the effective reproductive number of SARS-CoV-2. _eLife_ **11**, e71345 (2022).

2. Huisman, J. S. _et al._ Wastewater-Based Estimation of the Effective Reproductive Number of SARS-CoV-2. _Environmental Health Perspectives_ **130**, 057011 (2022).

3. Scire, J. _et al._ estimateR: an R package to estimate and monitor the effective reproductive number. _BMC Bioinformatics_ **24**, 310 (2023).

4. Lison, A. EpiSewer: Estimate Reproduction Numbers from Wastewater Measurements. Zenodo (2024).

5. Jones, D. L. _et al._ Shedding of SARS-CoV-2 in feces and urine and its potential role in person-to-person transmission and the environment-based spread of COVID-19. _Science of The Total Environment_ **749**, 141364 (2020).

6. Arts, P. J. _et al._ Longitudinal and quantitative fecal shedding dynamics of SARS-CoV-2, pepper mild mottle virus, and crAssphage. _mSphere_ **8**, e00132-23 (2023).

7. Nash, R. K., Bhatt, S., Cori, A. & Nouvellet, P. Estimating the epidemic reproduction number from temporally aggregated incidence data: A statistical modelling approach and software tool. _PLOS Computational Biology_ **19**, e1011439 (2023).

Reviewer #6: (See attached PDF for a full evaluation.)

The authors presented an R software package, which implements statistical methods to estimate the actual number of new infections using the number of reported cases or the wastewater data. It is important to note that the package allows the input data to be sampled by a period higher then one day (e.g., aggregated weekly data is also acceptable). Still, the output is a daily time series, which allows to estimate the effective reproduction number using the already existing tool, EpiEstim. To estimate the hidden time-series (i.e., the unknown input) from the measured output, the Authors applied a deconvolution using an existing Richardson-Lucy implementation. As far as I know, this technique is equivalent to a dynamic inversion, which was already used to infer the effective reproduction number.

Although I cannot detect any scientific contribution in this manuscript, the ``attached'' R package may be useful for a certain community (e.g., public health practitioners), and the software description in the main body is clear and didactic.

The manuscript has therefore a raison d'être, possibly not in such a high impact journal (but the Editor is the final judge on that).

Anyway, the Authors need to be better justify why their software tool is preferable or more convenient compared to other existing packages.

6. PLOS authors have the option to publish the peer review history of their article (what does this mean?). If published, this will include your full peer review and any attached files.

Reviewer #1: **Yes: **rewan abdelaziz

Reviewer #2: No

Reviewer #3: No

Reviewer #4: **Yes: **Tarek Abd El-Hafeez

Reviewer #5: **Yes: **Adrian Lison

Reviewer #6: **Yes: **Péter Polcz (Pázmány Péter Catholic University Faculty of Information Technology and Bionics)

---

## [Decision Letter · Decision Letter 1]

10 May 2024

PONE-D-24-01907R1ern: an R package to estimate the effective reproduction number using clinical and wastewater surveillance dataPLOS ONE

Dear Dr. Champredon,

Thank you for submitting your manuscript to PLOS ONE. After careful consideration, we feel that it has merit but does not fully meet PLOS ONE’s publication criteria as it currently stands. Therefore, we invite you to submit a revised version of the manuscript that addresses the points raised during the review process.

We look forward to receiving your revised manuscript.

Kind regards,

Salim Heddam

Academic Editor

PLOS ONE

Journal Requirements:

Additional Editor Comments:

Reviewer 5#:Thank you for the thorough revision and for addressing the comments. I have two remaining remarks, which however do not necessitate another review round in my opinion.

First, thank you for answering my question with regard to accounting for uncertainty of the Rt estimates towards the present. I understand the current deconvolution method is not able to account for this uncertainty. But I think it is very important to highlight this limitation more strongly in the manuscript, i.e. in the discussion, but also to make users aware of this in the package documentation etc. Otherwise, the use of the package could lead to wrong conclusions when using it for real-time surveillance. The uncertainty towards the present because of partial information is an inherent characteristic of the data and means that we can only get accurate Rt estimates with a certain delay. If this is not reflected in the uncertainty information provided by the method, users should be made aware of this limitation.

Second, you describe that to pool the uncertainty of the Rt estimates, you get the mean and quantiles for each realization and then compute means or quantiles of these across realizations. This seems rather approximate to me, and I wonder why you don't just draw e.g. 100 Rt samples from the posterior Gamma distribution as described by Cori et al. (you can use the estimated mean and cv from EpiEstim) for each realization, then combine all the draws across your realizations and compute the mean, empirical quantiles etc. on this pooled posterior sample. This should also be very fast and more accurate.

Reviewers' comments:

Reviewer's Responses to Questions

**Comments to the Author**

1. If the authors have adequately addressed your comments raised in a previous round of review and you feel that this manuscript is now acceptable for publication, you may indicate that here to bypass the “Comments to the Author” section, enter your conflict of interest statement in the “Confidential to Editor” section, and submit your "Accept" recommendation.

Reviewer #1: All comments have been addressed

Reviewer #2: All comments have been addressed

Reviewer #3: All comments have been addressed

Reviewer #4: All comments have been addressed

Reviewer #5: (No Response)

Reviewer #6: All comments have been addressed

2. Is the manuscript technically sound, and do the data support the conclusions?

Reviewer #1: Yes

Reviewer #2: Yes

Reviewer #3: Yes

Reviewer #4: Partly

Reviewer #5: Yes

Reviewer #6: Yes

3. Has the statistical analysis been performed appropriately and rigorously? 

Reviewer #1: (No Response)

Reviewer #2: Yes

Reviewer #3: Yes

Reviewer #4: Yes

Reviewer #5: Yes

Reviewer #6: N/A

4. Have the authors made all data underlying the findings in their manuscript fully available?

Reviewer #1: Yes

Reviewer #2: Yes

Reviewer #3: Yes

Reviewer #4: (No Response)

Reviewer #5: Yes

Reviewer #6: Yes

5. Is the manuscript presented in an intelligible fashion and written in standard English?

Reviewer #1: Yes

Reviewer #2: Yes

Reviewer #3: Yes

Reviewer #4: (No Response)

Reviewer #5: Yes

Reviewer #6: Yes

6. Review Comments to the Author

**Reviewer #1**: Thanks for your reply

**Reviewer #2**: (No Response)

**Reviewer #3:** (No Response)

**Reviewer #4: **An updated manuscript addressing previous comments and suggestions was evaluated positively. The updated submission demonstrates significant improvement and provides valuable insights relevant to the research community.

**Reviewer #5: **Thank you for the thorough revision and for addressing the comments. I have two remaining remarks, which however do not necessitate another review round in my opinion.

First, thank you for answering my question with regard to accounting for uncertainty of the Rt estimates towards the present. I understand the current deconvolution method is not able to account for this uncertainty. But I think it is very important to highlight this limitation more strongly in the manuscript, i.e. in the discussion, but also to make users aware of this in the package documentation etc. Otherwise, the use of the package could lead to wrong conclusions when using it for real-time surveillance. The uncertainty towards the present because of partial information is an inherent characteristic of the data and means that we can only get accurate Rt estimates with a certain delay. If this is not reflected in the uncertainty information provided by the method, users should be made aware of this limitation.

Second, you describe that to pool the uncertainty of the Rt estimates, you get the mean and quantiles for each realization and then compute means or quantiles of these across realizations. This seems rather approximate to me, and I wonder why you don't just draw e.g. 100 Rt samples from the posterior Gamma distribution as described by Cori et al. (you can use the estimated mean and cv from EpiEstim) for each realization, then combine all the draws across your realizations and compute the mean, empirical quantiles etc. on this pooled posterior sample. This should also be very fast and more accurate.

**Reviewer #6:** Upon perusing the Authors' response letter, I have become convinced of the manuscript's significant contribution to the scientific community. In the revised manuscript, the authors have carefully outlined the discipline to which they wish to contribute and how. Table 1 is very useful and informative.

7. PLOS authors have the option to publish the peer review history of their article (what does this mean?). If published, this will include your full peer review and any attached files.

Reviewer #1: No

Reviewer #2: No

Reviewer #3: No

Reviewer #4: **Yes: **Tarek Abd El-Hafeez

Reviewer #5: **Yes: **Adrian Lison

Reviewer #6: No

---

## [Author Response · Author response to Decision Letter 1]

28 May 2024

Response to Reviewers – Round 2

We would like to thank the reviewers for their time to read and comment on our revised manuscript. 

In response to Reviewer #5’s comments:

Thank you for the thorough revision and for addressing the comments. I have two remaining remarks, which however do not necessitate another review round in my opinion.

First, thank you for answering my question with regard to accounting for uncertainty of the Rt estimates towards the present. I understand the current deconvolution method is not able to account for this uncertainty. But I think it is very important to highlight this limitation more strongly in the manuscript, i.e. in the discussion, but also to make users aware of this in the package documentation etc. Otherwise, the use of the package could lead to wrong conclusions when using it for real-time surveillance. The uncertainty towards the present because of partial information is an inherent characteristic of the data and means that we can only get accurate Rt estimates with a certain delay. If this is not reflected in the uncertainty information provided by the method, users should be made aware of this limitation.

Response: Thank you for this suggestion. We have added a paragraph in the Discussion section to reflect this limitation.

Second, you describe that to pool the uncertainty of the Rt estimates, you get the mean and quantiles for each realization and then compute means or quantiles of these across realizations. This seems rather approximate to me, and I wonder why you don't just draw e.g. 100 Rt samples from the posterior Gamma distribution as described by Cori et al. (you can use the estimated mean and cv from EpiEstim) for each realization, then combine all the draws across your realizations and compute the mean, empirical quantiles etc. on this pooled posterior sample. This should also be very fast and more accurate.

Response: Thank you very much for this suggestion. Indeed, the way this was implemented was not statistically correct (we were not aware of the EpiEstim function “sample_posterior_R”). It is now implemented as suggested (link). A few tests showed that the numerical difference between both implementations was very small. The main text has been edited to reflect this change.

---

## [Decision Letter · Decision Letter 2]

2 Jun 2024

ern: an R package to estimate the effective reproduction number using clinical and wastewater surveillance data

PONE-D-24-01907R2

Dear Dr. Champredon

We’re pleased to inform you that your manuscript has been judged scientifically suitable for publication and will be formally accepted for publication once it meets all outstanding technical requirements.

Kind regards,

Salim Heddam

Academic Editor

PLOS ONE

Additional Editor Comments (optional):

Reviewers' comments:

Reviewer's Responses to Questions

**Comments to the Author**

1. If the authors have adequately addressed your comments raised in a previous round of review and you feel that this manuscript is now acceptable for publication, you may indicate that here to bypass the “Comments to the Author” section, enter your conflict of interest statement in the “Confidential to Editor” section, and submit your "Accept" recommendation.

Reviewer #5: All comments have been addressed

2. Is the manuscript technically sound, and do the data support the conclusions?

Reviewer #5: Yes

3. Has the statistical analysis been performed appropriately and rigorously? 

Reviewer #5: Yes

4. Have the authors made all data underlying the findings in their manuscript fully available?

Reviewer #5: Yes

5. Is the manuscript presented in an intelligible fashion and written in standard English?

Reviewer #5: Yes

6. Review Comments to the Author

Reviewer #5: (No Response)

7. PLOS authors have the option to publish the peer review history of their article (what does this mean?). If published, this will include your full peer review and any attached files.

Reviewer #5: **Yes: **Adrian Lison

---

## [Editor Report · Acceptance letter]

7 Jun 2024

PONE-D-24-01907R2 

PLOS ONE

Dear Dr. Champredon, 

I'm pleased to inform you that your manuscript has been deemed suitable for publication in PLOS ONE. Congratulations! Your manuscript is now being handed over to our production team.

Kind regards, 

on behalf of

Dr. Salim Heddam 

Academic Editor

PLOS ONE